# Differences in Water Dynamics between the Hydrated Chitin and Hydrated Chitosan Determined by Quasi-Elastic Neutron Scattering

**DOI:** 10.3390/bioengineering10050622

**Published:** 2023-05-22

**Authors:** Yuki Hirota, Taiki Tominaga, Takashi Kawabata, Yukinobu Kawakita, Yasumitsu Matsuo

**Affiliations:** 1Department of Life Science, Faculty of Science & Engineering, Setsunan University, Ikeda-Nakamachi, Neyagawa 572-8508, Osaka, Japan; takashi.kawabata@setsunan.ac.jp (T.K.); ymatsuo@lif.setsunan.ac.jp (Y.M.); 2Neutron Science and Technology Center, Comprehensive Research Organization for Science and Society (CROSS), Tokai, Nakagun 319-1106, Ibaraki, Japan; t_tominaga@cross.or.jp; 3Japan Proton Accelerator Research Complex (J-PARC) Center, Japan Atomic Energy Agency, Tokai, Nakagun 319-1195, Ibaraki, Japan

**Keywords:** biomaterial, fuel cell electrolyte, chitin, chitosan, hydration water dynamics, hydrogen atoms dynamics, proton conductivity, quasi-elastic neutron scattering (QENS), proton acceptors

## Abstract

Recently, it was reported that chitin and chitosan exhibited high-proton conductivity and function as an electrolyte in fuel cells. In particular, it is noteworthy that proton conductivity in the hydrated chitin becomes 30 times higher than that in the hydrated chitosan. Since higher proton conductivity is necessary for the fuel cell electrolyte, it is significantly important to clarify the key factor for the realization of higher proton conduction from a microscopic viewpoint for the future development of fuel cells. Therefore, we have measured proton dynamics in the hydrated chitin using quasi-elastic neutron scattering (QENS) from the microscopic viewpoint and compared the proton conduction mechanism between hydrated chitin and chitosan. QENS results exhibited that a part of hydrogen atoms and hydration water in chitin are mobile even at 238 K, and the mobile hydrogen atoms and their diffusion increase with increasing temperature. It was found that the diffusion constant of mobile protons is two times larger and that the residence time is two times faster in chitin than that in chitosan. In addition, it is revealed from the experimental results that the transition process of dissociable hydrogen atoms between chitin and chitosan is different. To realize proton conduction in the hydrated chitosan, the hydrogen atoms of the hydronium ions (H_3_O^+^) should be transferred to another hydration water. By contrast, in hydrated chitin, the hydrogen atoms can transfer directly to the proton acceptors of neighboring chitin. It is deduced that higher proton conductivity in the hydrated chitin compared with that in the hydrated chitosan is yielded by the difference of diffusion constant and the residence time by hydrogen-atom dynamics and the location and number of proton acceptors.

## 1. Introduction

Polysaccharide, one of the biomaterials, attracts attention as renewable resources. Polysaccharide is abundant in nature and is known as environmentally friendly biomaterials. In particular, chitin and chitosan are the second most abundant biopolymers in the world. Chitin is found in the exoskeletons (shells) of crustaceans such as shrimp and crabs, and insects, while chitosan is found in some fungi, diatoms, sponges, worms, and mollusks and is generally produced industrially by chemically deacetylating the chitin. Chitin has a hydroxyl group (-OH) and N-acetaldehyde (NHCOCH_3_), and chitosan has a hydroxyl group (-OH) and an amino group (-NH_2_), which are highly polar functional groups. Furthermore, these reactive groups can be easily functionally converted by chemical modification [1,2]. Their chemical structures are very similar, and they have excellent characteristics such as antibacterial, moisture retention, biocompatibility, safety, chelating, selective ion permeability, and low cost due to their natural origin [3,4]. Despite the fact that chitin and chitosan have a high versatility, several tons of them are still discarded each year. They can be easily formed into different shapes and are utilized in industries in various forms, such as fibers, hydrogels, beads, sponges, membranes, etc. Recently, chitin and chitosan have been used in cancer treatment as a vehicle for delivering cancer drugs to a specific site and in tissue/organ regeneration and repair [5,6]. Furthermore, it has been reported that the hydrated chitin and chitosan films in humidified conditions become proton conductors similar to other biomaterials [7,8,9,10,11,12,13,14,15], and that can be used as the fuel cell electrolyte where the chemical energy by hydrogen and oxygen is directly extracted as electric energy. Thus, chitin and chitosan are versatile and have the potential to play an important role in a wide range of industries, and are the most attractive biomaterials. The fuel cell is a key device for achieving a carbon-neutral society without CO_2_ emission [16,17]. Some problems for the currently-used fuel cell, such that electrolyte and platinum electrodes are expensive and cause significant environmental impact in production and disposal, can be solved by using biomaterials with low-cost and environmentally friendly properties.

Figure 1 shows the temperature dependence of ionic conductivities (*σ*_AC_) of hydrated chitin membranes with H_2_O (Milli-Q) or D_2_O (heavy water) described as “Chitin-H_2_O” and “Chitin-D_2_O” and chitosan membranes hydrated with H_2_O and D_2_O described as “Chitosan-H_2_O” and “Chitosan-D_2_O”. The temperature dependence of the ionic conductivity of Chitin-H_2_O and Chitin-D_2_O was measured using a precision LCR meter (Agilent E4980A) in the temperature range 230–280 K with the Chitin-H_2_O and Chitin-D_2_O membranes of 0.08 μm thickness, according to the same technique that determined that of Chitosan-H_2_O and Chitosan-D_2_O [18]. The ionic conductivities of Chitin-H_2_O and Chitosan-H_2_O increase as the temperature increase, and these values are higher than that of Chitin-D_2_O and Chitosan-D_2_O. Furthermore, the ionic conductivity of Chitin-H_2_O is ~30 times higher than that of Chitosan-H_2_O. Considering that deacetylation of chitin modifies the N-acetaldehyde to the amino group, it is expected that the acetyl group plays an important role in the increase of proton conductivity [14]. Regarding the electrolyte of a fuel cell, it is very important to have high proton conductivity, and clarifying the proton conduction mechanism from a microscopic viewpoint is necessary for the future development of fuel cells.

Recently, we have measured the hydrated chitosan from the microscopic viewpoint to reveal water dynamics via quasi-elastic neutron scattering (QENS) [18]. QENS provides quantitative observation of mono-particle dynamics information such as a self-diffusion and site-to-site jump distance by analyzing the momentum transfers (*Q*) and energy transfers (Δ*E* = Δ*ħω*) between an incident neutron and a neutron scattered from a nucleus [19]. The neutron energy and momentum transfer provide information on the time and space distributions of the particle. From the extremely large incoherent scattering cross section of hydrogen, the scattering spectra obtained from H_2_O-containing materials focus on self-diffusion dynamics of hydrogen atoms derived from the materials and water molecules interacting with the material, while the scattering spectra obtained from D_2_O-containing materials only focus on hydrogen atoms derived from the materials. Hydrated water dynamics has been evaluated by subtraction of the spectrum for the material hydrated with D_2_O from that hydrated with H_2_O, assuming no functional difference in the dynamics of the material between the D_2_O and H_2_O systems. The results of water dynamics in chitosan by QENS measurements showed that mobile hydration water exhibits jump diffusion motion and a diffusion constant increased from 1.33 × 10^−6^ cm^2^/s at 238 K to 1.34 × 10^−5^ cm^2^/s at 283 K [18]. Moreover, a part of the hydrogen atoms (i.e., mobile hydrogen atoms) attached to chitosan dissociates as the protons and undergoes jump diffuses. In addition, the amount of the mobile hydrogen well corresponds with the proportion of the mobile hydration water surrounding the dissociable hydrogen atoms in chitosan, which is observed at each temperature. From these results, it was found that the transition of protons between the hydration water molecule and hydrogen atoms of chitosan is significantly important to realize proton conduction in the hydrated chitosan.

In the present study, we have investigated water dynamics in the hydrated chitin by QENS measurement and revealed differences in proton conductivity between the hydrated chitin and chitosan from the microscopic viewpoint. The results in the present work will lead not only to the mechanism of proton conductivity in chitin and chitosan but to the development of the application of new environmentally friendly bio-electric devices such as fuel cells and hydrogen sensors. 

## 2. Samples and Experiments

### 2.1. Chitin Samples

A chitin 0.08 mm-thickness film was prepared by using a 20 g chitin slurry of 2 wt% chitin nanofiber (BiNFi-s Chitin, almost 100% purity) made by Sugino Machine Ltd. (Toyama, Japan). The chitin membranes, removed from water by suction filtration through a PTFE membrane filter from chitin slurry and shaped, were dried in a desiccator under a phosphorus pentoxide drying agent 2 days to prepare the dried chitin membranes, described as “Dry-Chitin”. In addition, the hydrated chitin membranes were prepared by immersing Dry-Chitin in H_2_O (Milli-Q) test tubes for 2 days, which is described as “Chitin^H^”. Furthermore, the hydrated chitin membranes with heavy water were also prepared by immersing Dry-Chitin in D_2_O test tubes for 2 days, which is described as “Chitin^D^”. Figure 2 shows their manufacturing process of chitin membrane samples.

### 2.2. QENS Measurements

QENS experiments for Chitin^H^ and Chitin^D^ were performed by using a time-of-flight (TOF) backscattering neutron spectrometer (DNA) [20,21,22,23] installed at the BL02 neutron port at the Materials and Life Science Experimental Facility (MLF) in the Japan Proton Accelerator Research Complex (J-PARC), Tokai, Ibaraki, Japan. In the experiment, the energy transfer from −500 μeV to 1500 μeV with 12 μeV energy resolution and the momentum transfer from 0.08 to 1.78 Å^−1^ have been surveyed, which corresponds to in time range and in space range. These time-space scales are suitable for thermal fluctuations and interatomic distance scales in polymers, which enable the microscopic observation of dynamics in the mono-particles via QENS measurements.

Chitin^H^ and Chitin^D^ prepared in dimensions of 4.5 cm × 4.5 cm × 0.08 mm achieved 5 water molecules per chitin monomer for the QENS measurements. These samples were set up in an aluminum cylindrical cell as the procedure for chitosan QENS measurements [18]. The temperature dependence of the hydration water states of Chitin^H^ and Chitin^D^ were measured using a sample changer PEACE with a function of successive measurement for three samples [24]. The QENS spectra were analyzed by DAVE [25].

Since the chitin monomer includes 13 hydrogen atoms and the incoherent neutron scattering cross-sections of the hydrogen atoms in the Chitin^D^ sample are ~10 times larger than that of the coherence neutron scattering cross-section, the scattering spectra for Chitin^D^, which are described as “*S*_atom_ (*Q*, *ω*)”, was interpreted as the mono-particle dynamics of hydrogen atoms bonded to chitin. On the other hand, the scattering spectra of Chitin^H^ described as “*S*_all_ (*Q*, *ω*)” was associated with contributions both from the hydration of water and the hydrogen atoms in chitin. *S*_water_ (*Q*, *ω*) was obtained by deducting *S*_atom_ (*Q*, *ω*) from *S*_all_ (*Q*, *ω*) to distinguish only the hydration water dynamics in the hydrated chitin.

## 3. Results

### 3.1. Quasi-Elastic Neutron Scattering Signals in Chitin; S_water_ (Q, ω)

Figure 3 shows the QENS spectra of *S*_water_ (*Q*, *ω*) in chitin from 238 K to 283 K at *Q* = 1.0 Å^−1^. Even at 238 K, where bulk water is generally frozen, the spectra were appreciably wider than the resolution. This result means that a part of the hydration water is mobile. The spectral width gradually increased in their width as the temperature increased until 268 K and significantly broadened at 283 K.

### 3.2. Quasi-Elastic Neutron Scattering Signals in Chitin; S_atom_ (Q, ω)

Figure 4 shows the QENS spectra of *S*_atom_ (*Q*, *ω*) from 238 K to 283 K at *Q* = 1.0 Å^−1^. Their overall trend that the spectral width gradually larger as the temperature increased was close to that of *S*_water_ (*Q*, *ω*).

## 4. Results of QENS Spectrum Analysis

### 4.1. S_water_ (Q, ω): Hydration Water Dynamics in the Hydrated Chitin

As shown in Figure 3, *S*_water_ (*Q*, *ω*) spectra consist of at least a sharp and broad component. Therefore, to evaluate the hydration water dynamics in the hydrated chitin, these spectra were initially fitted by the sum of a delta and Lorentz function convolved with the energy resolution of the instrument, as the following Equation (1):*S*_water_ (*Q*, *ω*) = {*X*_immobile_
*δ*(*ω*) + *Y*_mobile_
*L*(Γ_slow_, *ω*)} ⊗ *R* (*Q*, *ω*) + BG(1)
where *δ*, *L*, Γ_slow_, ⊗, *R*, and BG are the delta function, Lorentz function, a half-width at half maximum (HWHM) of the Lorentz function, the convolution operator, the instrument energy resolution, and instrumental background, respectively. *X*_immobile_ and *Y*_mobile_ are the coefficients of the corresponding components. The *δ*(*ω*) and *L*(Γ_slow_, *ω*) represent the elastic component of the immobile and QENS component of the slow mobile hydration water, respectively. 

As shown in Figure 5, the *S*_water_ (*Q*, *ω*) in chitin from 238 K to 268 K can be reproduced by Equation (1) at all *Q*. However, reproducing *S*_water_ (*Q*, *ω*) in chitin at 283 K required an additional Lorentz function. *L*(Γ_fast_, *ω*) represents the fast mobile hydration water, and *S*_water_ (*Q*, *ω*) at 283 K is reproduced by Equation (2):*S*_water_ (*Q*, *ω*) = {*X*_immobile_
*δ*(*ω*) + *Y*_mobile_
*L*(Γ_slow_, *ω*) + *Z*_mobile_
*L*(Γ_fast_, *ω*)} ⊗ *R* (*Q*, *ω*) + BG(2)
where *Z*_mobile_ is the coefficient of the Lorentz function with Γ_fast_. 

Figure 6 shows the *Q*^2^-dependence of Γ_slow_ and Γ_fast_ in chitin from 238 K to 283 K. The Γ_slow_ and Γ_fast_ gradually increased as *Q*^2^ increased, while the amount of increase was smaller on the high-*Q*^2^ side. As the temperature increased, the Γ_slow_ increased with keeping similar *Q*^2^-dependence. The Γ_fast_ was more than ∼3.0 times larger than the Γ_slow_ at 283 K.

The *Q*^2^-dependences of Γ_slow_ and Γ_fast_ in chitin were analyzed in Equation (3) according to the jump-diffusion model [19].
(3)$$\Gamma \left(Q\right)=\frac{DQ^{2}}{1+\left(DQ^{2}\tau \right)}\;\;$$
where *τ* and *D* represent the residence time and the self-diffusion coefficient, respectively. The jump-diffusion model represents the process of the single particle staying at a certain position for *τ* seconds and then jumping a distance *l*. The *l* is estimated by *l* = $\sqrt{6D\tau }$ by *D* and *τ*.

The jump-diffusion model reproduces the *Q*^2^-dependence of Γ_slow_ and Γ_fast_ in chitin well, as black broken lines in Figure 6, and then, *D*, *τ*, and *l* for the slow and fast mobile hydration water in chitin at different temperatures were evaluated.

Figure 7 shows the temperature dependence of *D* and *τ* for the fast and the slow mobile hydration water in chitin and chitosan [18], and the results for the slow mobile hydration water in chitin obey the Arrhenius law with the activation energies 0.28 eV. The *D* and *τ* for the fast mobile hydration water in chitin were ~1.5 times larger and ~6.0 times faster than those of the slow mobile hydration water in chitin, respectively. As shown in Figure 8, *l* for the slow mobile hydration water in chitin estimated was almost constant to be 2.5 ± 0.01 Å. *l* for the fast mobile hydration water in chitin at 283 K is 1.2 ± 0.05 Å and close to that for free water [26]. There were no significant differences between chitin and chitosan for *D*, *τ*, and *l* of the slow mobile hydration water.

Figure 9 shows the temperature dependence of the fractions (*F%*) of three types for hydration water (i.e., the immobile, slow mobile, and fast mobile hydration water) of different mobility for the hydration water in chitin and chitosan [18]. The coefficients of the delta (i.e., *X*_immobile_) and Lorenz functions (i.e., *Y*_mobile_ and Z_mobile_) obtained by fitting Equations (1) and (2) are proportionate to the quantity of the respective hydration water components. These fractions, *F*%, can be calculated from the following Equations (4)–(6).
(4)$$F_{\mathrm{immobile}}=\left(\frac{X_{immobile}}{X_{immobile}+Y_{\mathrm{mobile}\;}+Z_{\mathrm{mobile}\;}}\right)\;\times \;100,$$
(5)$$F_{\mathrm{slow}\;\mathrm{mobile}}=\left(\frac{Y_{\mathrm{mobile}}}{X_{immobile}+Y_{\mathrm{mobile}\;}+Z_{\mathrm{mobile}\;}}\right)\;\times \;100,$$
(6)$$F_{\mathrm{fast}\;\mathrm{mobile}}=\left(\frac{Z_{\mathrm{mobile}}}{X_{immobile}+Y_{\mathrm{mobile}\;}+Z_{\mathrm{mobile}\;}}\right)\;\times \;100,$$

As the temperature increased, *F*% of the slow mobile hydration water in chitin increased (i.e., 3.2% at 238 K, 5.8% at 253 K, 13.8% at 268 K, and 49.6% at 283 K), and the fast mobile hydration water only observed at 283 K (i.e., 41.4%). Notably, the mobile hydration water in chitin was also observed even below 273 K, where bulk is generally frozen. The trend of increasing mobile hydration water in chitin as the temperature increases is similar to that of chitosan.

### 4.2. S_atom_ (Q, ω): Hydrogen Atoms Dynamics in Chitin

Since the sharp and broad components are also observed in the *S*_atom_ (*Q*, *ω*) spectra in Figure 4, the *S*_atom_ (*Q*, *ω*) spectra were fitted by Equation (7) to separate the immobile and mobile hydrogen atoms in chitin, as the following Equation (7):*S*_atom_ (*Q*, *ω*) = { *X*′ _immobile_
*δ*(*ω*) + *Y*′_mobile_
*L*(Γ_hydrogen atom_, *ω*)} ⊗ *R* (*Q*, *ω*) + BG(7)

As shown in Figure 10, the *S*_atom_ (*Q*, *ω*)s in chitin at all temperatures can be reproduced well by Equation (7).

Figure 11 shows the *Q*^2^-dependence of Γ_hydrogen atoms_ in chitin obtained at different temperatures. The Γ_hydrogen atom_ continuously increased as *Q*^2^ increased. Considering that a QENS response of *S*_atom_ (*Q*, *ω*) is the mono-particle dynamics of hydrogen atoms in chitin, these results indicate that a part of the hydrogen atoms in chitin exhibits diffusion behavior, and the amount of diffusion increased as the temperature increased.

It should be noted here that the chitin monomer has 13 hydrogen atoms, and the hydroxyl and the amino groups of N-acetaldehyde groups (i.e., N-H) groups dissociate more easily as protons than other C-H bonds. In fact, the Γ_hydrogen atom_ can be well reproduced by the jump-diffusion model Equation (3) [19], as shown by the black broken lines in Figure 11. 

The obtained *D* and *τ* for the mobile hydrogen atoms in chitin were plotted in the Arrhenius plots in Figure 12. These results reasonably lie on a straight line, as shown in Figure 12a,b. The activation energies of *D* and *τ* for the mobile hydrogen atoms in chitin obtained from the Arrhenius Equation were estimated to be 0.28 eV, and these values were consistent with the activation energies for the slow mobile hydration water in the hydrated chitin.

As shown in Figure 13, *l* for the mobile hydrogen atom in chitin estimated by *l* = $\sqrt{6D\tau }$ was almost constant to be 2.1 ± 0.01 Å in the temperature range from 238 K to 283 K. The jump distance in chitin was the same as that of chitosan. 

Figure 14 shows the temperature dependence of the fractions (*F%*) of two types for hydrogen atoms (i.e., the immobile, mobile hydrogen atoms) of different mobility in chitin and chitosan. The temperature dependence of fractions (*F*%) for the hydrogen atoms in chitin was determined using Equations (4) and (5), as same in the case of S_water_ (*Q*, *ω*). A small amount of mobile hydrogen atoms was observed even at 238 K, and it gradually increased as the temperature increased. The overall trend in the temperature dependence of mobile hydrogen atoms in chitin was similar to that of the slow mobile hydration water in chitin and that of mobile hydrogen atoms in chitosan.

## 5. Discussion

The main objective of this paper is to understand the proton conduction mechanism of hydrated chitin by investigating the relationship between the hydration water in chitin and the hydrogen atoms from the microscopic viewpoint and to determine the differences in proton conduction between the hydrated chitin and chitosan. The analysis results of *S*_atom_ (*Q*, *ω*) showed that a part of the hydrogen atoms in chitin was mobile even at 238 K and dissociated from chitin as protons. In addition, the analysis results of *S*_water_ (*Q*, *ω*) also showed a part of the hydration water in chitin was mobile even at 238 K, where bulk water is frozen, and had jump-diffusive obeying the jump-diffusion model [19]. The fraction of the slow mobile hydration water and the mobile hydrogen atoms in chitin increased as the temperature increased. These results are similar to trends of the results reported in the QENS experiments for the hydrated chitosan [18].

However, a quantitatively significant difference between the hydrogen atoms dynamics of chitin and chitosan was shown. More specifically, *D* of the mobile hydrogen atoms in chitin was two times larger, and *τ* was two times faster than those in chitosan. These results suggest that the difference in ionic conductivity between chitin and chitosan is due to *D* and *τ*.

Regarding the mobile hydrogen atom and slow mobile hydration water dynamics in chitin, *D* of the mobile hydrogen atom was 1.5 times larger, *τ* was two times faster, and *l* was 0.4 Å shorter than those of the slow mobile hydration. However, the activation energy estimated from these *D* and *τ* in the mobile hydrogen atoms and the slow mobile hydration water dynamics were consistent with 0.28 eV (cf. Figure 7 and Figure 12), suggesting that these dynamics are closely related.

Two conduction mechanisms of proton transport are generally well known, namely the Grotthuss and the Vehicle mechanism [27]. The Grotthuss mechanism is the proton hopping transfer within the hydrogen bonding network of water molecules. It is assumed that protons transfer around water molecules to form hydronium ions (H_3_O^+^) in the cluster and simultaneously break hydrogen bonds and subsequent rearrangement between the neighborhood water molecules. On the other hand, the vehicle mechanism is carried out by self-diffusion of proton-producing species. In a hydrated polymer electrolyte, the QENS results support that the water dynamics mechanism in Nafion is a diffusion mechanism in which H_3_O^+^ moves rather than a concerted jump motion in which H_3_O^+^ continuously builds up and disappears [28,29]. In addition, it is considered that the hydrated polymer electrolyte undergoes formation of H_3_O^+^ via a slow jump mode with a characteristic time from 500 ps to 150 ps and long-range diffusion of water molecules within the confinement region via a fast jump mode with a characteristic time from 8.0 ps to 2.5 ps [28,29]. The residence times of mobile hydrogen atoms in chitin ranged from 0.26 ps to 3.1 ps is much faster than these motions observed in the hydrated polymer electrolyte. Furthermore, these conduction mechanisms can be distinguished by their activation energies, and the proton conduction process by the Grotthuss mechanism generally requires activation energy ≤ 0.4 eV, while the vehicle mechanism requires a larger energy ≥ 0.4 eV because it transports a mass larger than the proton [27,30,31]. Since the activation energies for the hydrogen atoms and hydration water dynamics in chitin were below 0.4 eV, it is clear that proton conduction processes in the hydrated chitin are the Grotthuss mechanism.

In the proton conduction process of the hydrated chitosan, the hydrogen atoms of the hydroxyl and amino groups in chitosan dissociate as protons and diffuse into the neighboring hydration water (i.e., fast water or slow mobile water) to form H_3_O^+^. Subsequently, the mobile hydration water assists in the proton transfer between H_3_O^+^ and additional hydration water molecules, thereby realizing proton conduction. However, Chitosan-D_2_O does not exhibit a significant increase in ionic conductivity as temperature increases. (cf. Figure 1). Therefore, it is speculated that hydrogen atoms are unable to transfer to D_2_O hydration continuously. This means that the hydrogen atom of H_3_O^+^ would be returned to the chitosan when a proton acceptor is D_2_O. However, the ionic conductivity in Chitin-D_2_O increased as the temperature increased and exhibited high ionic conductivity (cf. Figure 1). Therefore, it is speculated that Chitin^D^ and Chitosan^D^ are different in the proton conduction process. For this reason, in the proton conduction process in chitin based on the above consideration, it is possible that the hydrogen atoms derived from chitin dissociate and jump to the neighboring hydration water molecules to form H_3_O^+^, and then the hydrogen atom of H_3_O^+^ directly transfers to the neighboring chitin. In order to determine a suitable model for the process leading to proton conduction in the hydrated chitin, the model for the relationship between the mobile hydration water and mobile hydrogen atoms was formed based on the following assumptions. (1) Among the thirteen hydrogen atoms per chitin monomer, there are enough mobile hydration waters around three dissociable hydrogen atoms (i.e., the hydroxyl and the amino groups of N-acetaldehyde groups) at 283 K, and these hydrogen atoms preferentially transfer to the slow mobile hydration water; (2) The dissociable hydrogen atoms equivalently transfer to the slow mobile hydration water to form H_3_O^+^; (3) The hydrogen atom of H_3_O^+^ can transfer directly to the proton acceptor of neighboring chitin. Thus the following equation was formulated,
(8)$$\frac{Y_{mobile}^{\prime }}{Y_{mobile}^{\prime }+X_{immobile}^{\prime }}=\frac{Y_{mobile}\;at\;each\;temperature}{\left(Y_{mobile}+Z_{mobile}\right)\;at\;283K}\times \frac{3}{13}\times \alpha$$
where *α* is the number of proton acceptors of neighboring chitin. Figure 15 shows the results of the theoretical equation when 3 is substituted for *α* and the fraction of the mobile hydrogen atoms in chitin (cf. Figure 14). The theoretical formula substituting *α* = 3 and the experimental data are equivalent, which indicates that the hydrogen atoms in chitin transfer to the proton acceptors of neighboring chitin via H_3_O^+^. These results indicate that the proton conduction process in the hydrated chitin, in addition to the proton transfer process between H_3_O^+^ and another hydration water observed in the hydrated chitosan, involves the preferential transfer of dissociable hydrogen atoms in chitin as protons to the slow mobile hydrogen water to form H_3_O^+^, and then the hydrogen atoms of H_3_O^+^ transfer to three proton acceptors of neighboring chitin.

These results indicate that the hydrogen atoms and hydration water are involved in the proton conduction of the hydrated chitin in the water dynamics observed in QENS, as schematically shown in Figure 16. In addition, it is known that the difference in proton conduction between the hydrated chitin and chitosan is attributed to the hydrogen atom dynamics. More in detail, the proton conduction process in the hydrated chitin, the dissociable hydrogen atoms (i.e., the hydroxyl and the amino groups of N-acetaldehyde groups) in chitin preferentially transfer as protons to the slow mobile hydration water located 2.1 Å in the distance to form H_3_O^+^, and then the hydrogen atom of H_3_O^+^ transfers to another mobile hydration water or the proton acceptors of neighboring chitin, thereby realizing proton conduction. Therefore, it is revealed that the difference in proton conductivity between the hydrated chitin and chitosan is determined by diffusion and residence time of hydrogen atom dynamics and the presence of proton acceptors in the distance that the hydrogen atoms of H_3_O^+^ can transfer and the number of proton acceptors for the hydrogen atoms of H_3_O^+^. We expect that the N-acetaldehyde groups in chitin play a role in controlling the location of the hydration water in order to ensure that dissociative hydrogen atoms can transfer directly to the proton acceptors of neighboring chitin.

## 6. Conclusions

We reported the results of neutron quasi-elastic scattering on Chitin^H^ and Chitin^D^ samples to reveal the proton conduction process in the hydrated chitin and differences in proton conductivity between the hydrated chitin and the hydrated chitosan from the microscopic viewpoint.

QENS results in chitin showed that a part of the hydration water and hydrogen atoms in chitin could be mobile even at 238 K, and the amount of mobility and diffusion increased as the temperature increased, which was qualitatively similar to the QENS results for chitosan. However, *D* of the mobile hydrogen atoms in chitin was ~two times larger, and *τ* was ~two times faster than those in chitosan, indicating a quantitative difference between chitin and chitosan. In the proton conduction process of the hydrated chitosan, the easily dissociable hydrogen atom (i.e., the hydroxyl and amino groups) dissociate as protons and transfer to the neighboring hydration water (i.e., the fast or slow mobile hydration water) to form H_3_O^+^, and then hydrogen atom of H_3_O^+^ transfers to an additional hydration water molecule to realize proton conduction. However, the ionic conductivity of Chitosan^D^ did not significantly increase as Chitin^D^. From these results, it is clear that the proton conduction process between chitin and chitosan is different. From the results of QENS measurements in chitin, the experimentally determined fraction of mobile hydrogen atoms was well correlated with the proportion that the dissociable hydrogen atoms (i.e., the hydroxyl and the amino groups of N-acetaldehyde groups) in chitin preferentially transfer to the slow mobile hydrogen atoms and then to the three proton acceptors of neighboring chitin. These results indicate that the proton conduction process of the hydrated chitin is not only the proton transfer process observed in the hydrated chitosan but also the dissociable hydrogen atoms in the chitin preferentially transfer as protons to the slow mobile hydrogen water to form H_3_O^+^, and then the hydrogen atoms of H_3_O^+^ transfer to 3 proton acceptors of neighboring chitin. Therefore, it is revealed that the difference in proton conductivity between the hydrated chitin and chitosan is yielded by the differences in *D* and *τ* determined by hydrogen-atom dynamics and the location and number of proton acceptors. 

Lastly, we concluded that the QENS measurements in chitin provide insight into the proton conduction process in the hydrated chitin by distinguishing between hydration water and hydrogen atom dynamics and understanding the ionic conductivity differences between chitin and chitosan from the microscopic viewpoint. These results are useful for research on the application of biopolymer chitin and chitosan, an environmentally friendly material, as a fuel cell electrolyte and for the potential functionality of chitin and chitosan for the creation of new products.

## Figures and Tables

**Figure 1 bioengineering-10-00622-f001:**
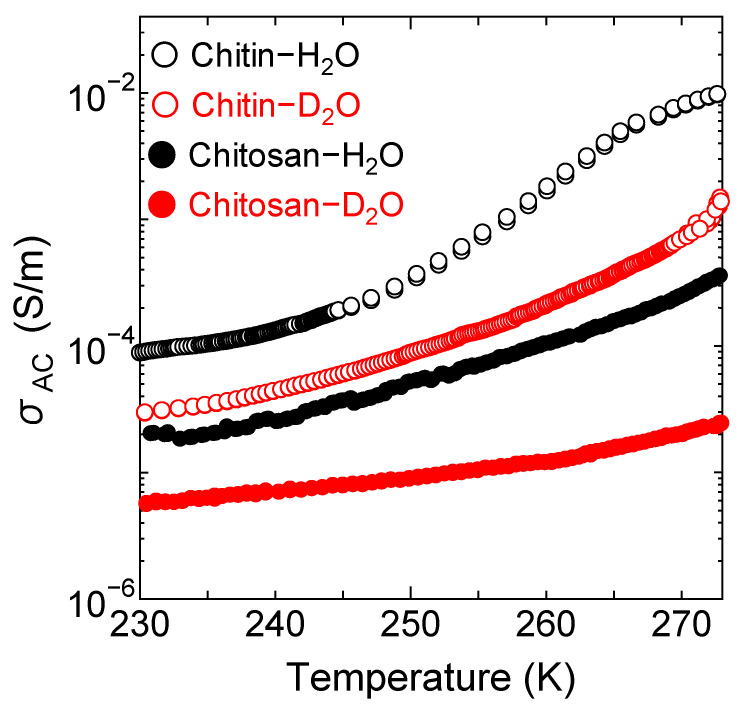
Temperature dependence of ionic conductivities (*σ*_AC_) of the hydrated chitin and chitosan membranes. The hydrated chitin with H_2_O (Milli-Q) or D_2_O (heavy water) is described as Chitin-H_2_O (black open circles) and Chitin-D_2_O (red open circles), respectively. The hydrated chitosan with H_2_O or D_2_O is described as Chitosan-H_2_O (black circles) and Chitosan-D_2_O (red circles), respectively. Data in chitosan is adapted with permission from Ref. [18].

**Figure 2 bioengineering-10-00622-f002:**
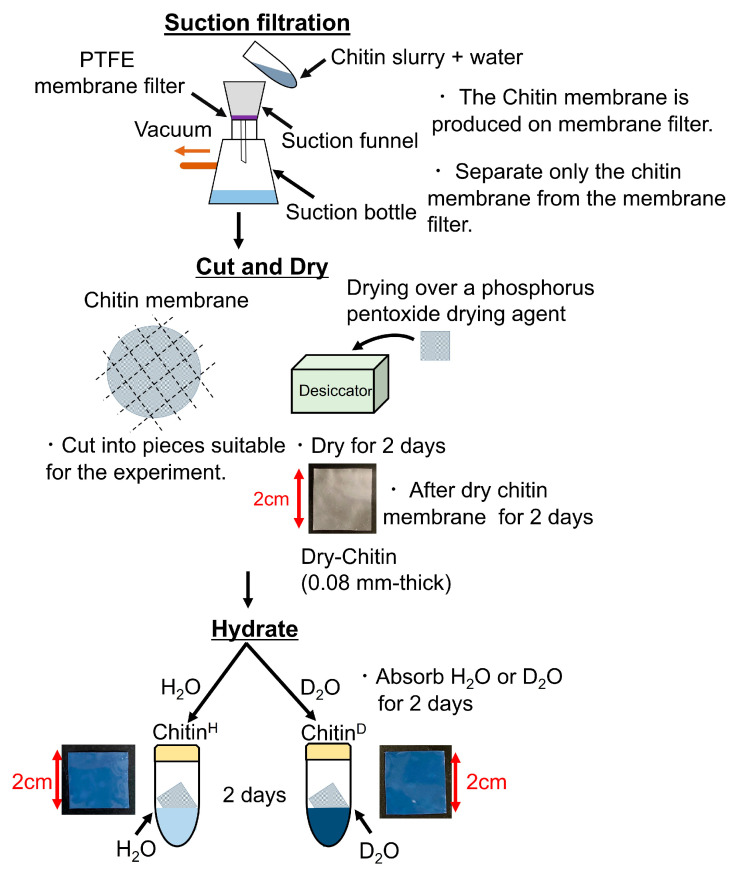
The manufacturing process of chitin membrane samples.

**Figure 3 bioengineering-10-00622-f003:**
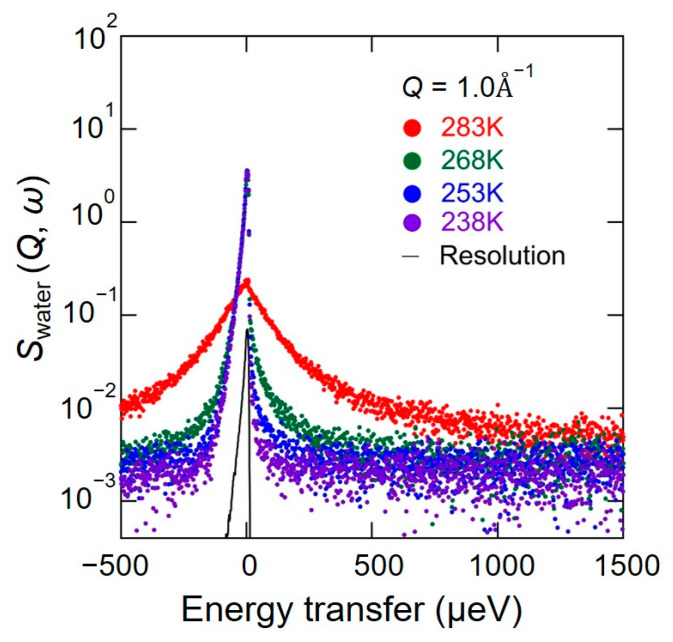
QENS spectra of *S*_water_ (*Q*, *ω*) in chitin from 238 K to 283 K at a *Q* value of 1.0 Å^−1^ obtained by deducting *S*_atom_ (*Q*, *ω*) from *S*_all_ (*Q*, *ω*) where *S*_all_ (*Q*, *ω*) and *S*_atom_ (*Q*, *ω*) are the dynamic structure factors of the hydrated chitin with H_2_O and with D_2_O, respectively. *S*_water_ (*Q*, *ω*) focuses on the hydration water dynamics in the hydrated chitin. The color-painted circles (i.e., 238 K (purple), 253 K (blue), 268 K (green), and 283 K (red)) are experimental data, and the black curve is an instrumental resolution spectrum obtained by QENS measurement of vanadium. These experimental data are observed in an energy window from −500 μeV to 1500 μeV with a resolution of 12 μeV.

**Figure 4 bioengineering-10-00622-f004:**
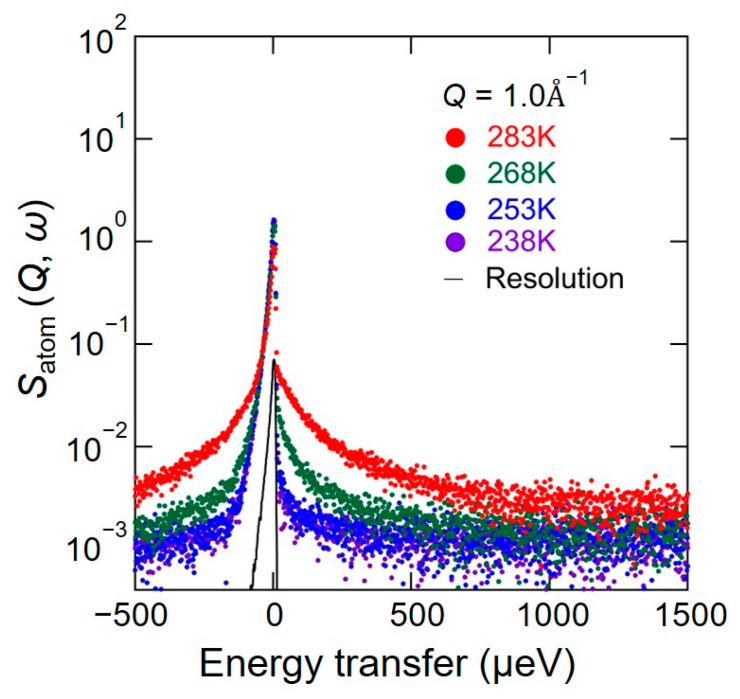
QENS spectra of *S*_atom_ (*Q*, *ω*) in chitin from 238 K to 283 K at a *Q* value of 1.0 Å^−1^. *S*_atom_ (*Q*, *ω*) is the dynamic structure factor of the hydrated chitin with D_2_O and represents the hydrogen atoms dynamics in chitin. The color-painted circles (i.e., 238 K (purple), 253 K (blue), 268 K (green), and 283 K (red)) are experimental data, and the black curve is an instrumental resolution spectrum obtained by QENS measurement of vanadium. These experimental data are observed in an energy window from −500 μeV to 1500 μeV with a resolution of 12 μeV.

**Figure 5 bioengineering-10-00622-f005:**
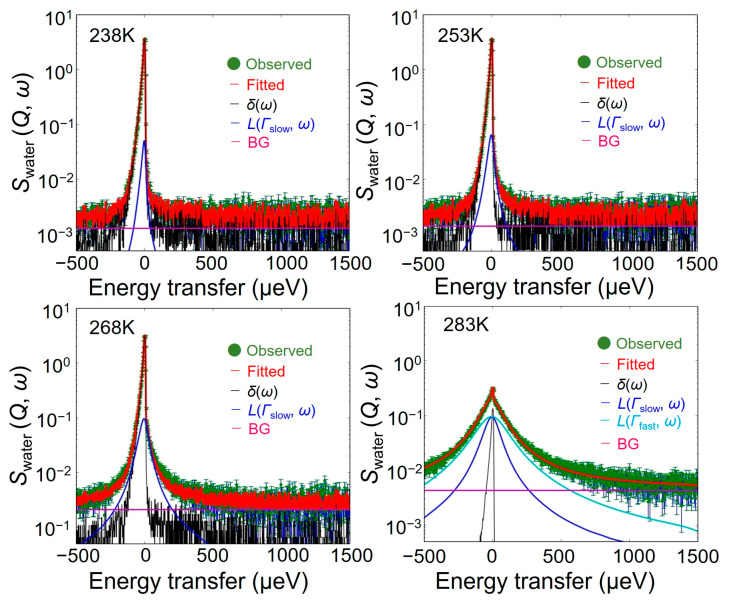
QENS spectrum of *S*_water_ (*Q*, *ω*) in chitin and fitting results from 238 K to 283 K at constant-*Q* = 1.0 Å^−1^. The green point and red line represent the experimental data and the results of fitting to Equations (1) and (2). The black, blue, and light blue lines are *δ*(*ω*), *L*(Γ_slow_, *ω*), and *L*(Γ_fast_, *ω*) in Equations (1) and (2), respectively. These are convoluted with the instrument resolution. The magenta line is the instrumental background.

**Figure 6 bioengineering-10-00622-f006:**
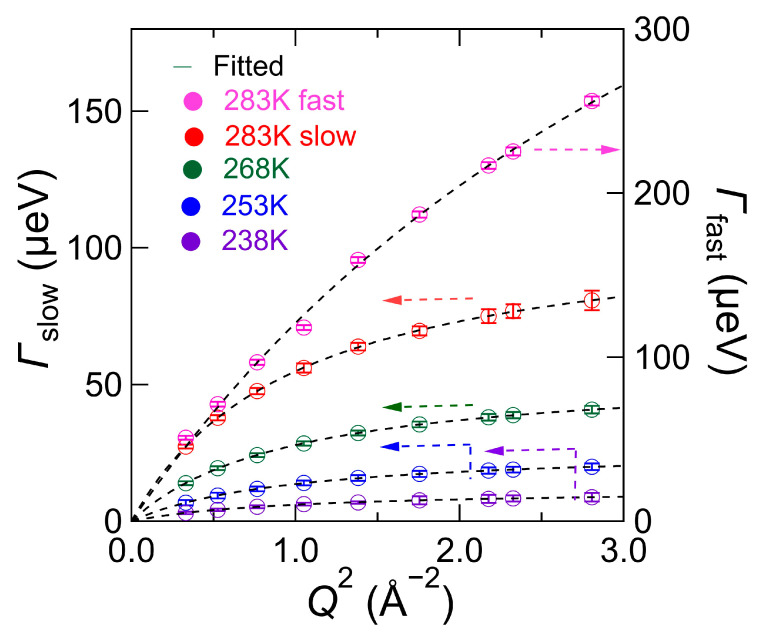
The *Q*^2^-dependence of half-width at half-maximum (HWHM) values of the quasi-elastic components (i.e., Γ_slow_ and Γ_fast_) for *S*_water_ (*Q*, *ω*) in chitin obtained by QENS profile fitted by Equations (1) and (2). The Γ_slow_ and Γ_fast_ are referred to as the left axis and the right axis. The experimental results at 238 K (purple), 253 K (blue), 268 K (green), and 283 K (red) for Γ_slow_ and 283 K (pink) for Γ_fast_ are fitted by the jump-diffusion model (black broken line) [19].

**Figure 7 bioengineering-10-00622-f007:**
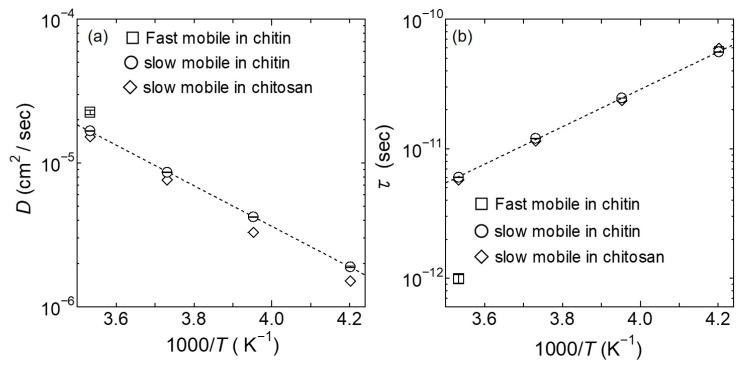
The temperature dependence of (**a**) *D* and (**b**) *τ* for the fast and slow mobile hydration water as a function of inverse temperature. The open circles and square are the slow and fast mobile hydration water in chitin, respectively. The open rhombuses are the slow mobile hydration water in chitosan [18]. The black dotted lines on (**a**) *D* and (**b**) *τ* represent the Arrhenius fit to the data of the slow mobile hydration water in chitin. The errors estimated from the fitting procedure are shown in a bar graph on each mark. Data in chitosan is adapted with permission from Ref. [18].

**Figure 8 bioengineering-10-00622-f008:**
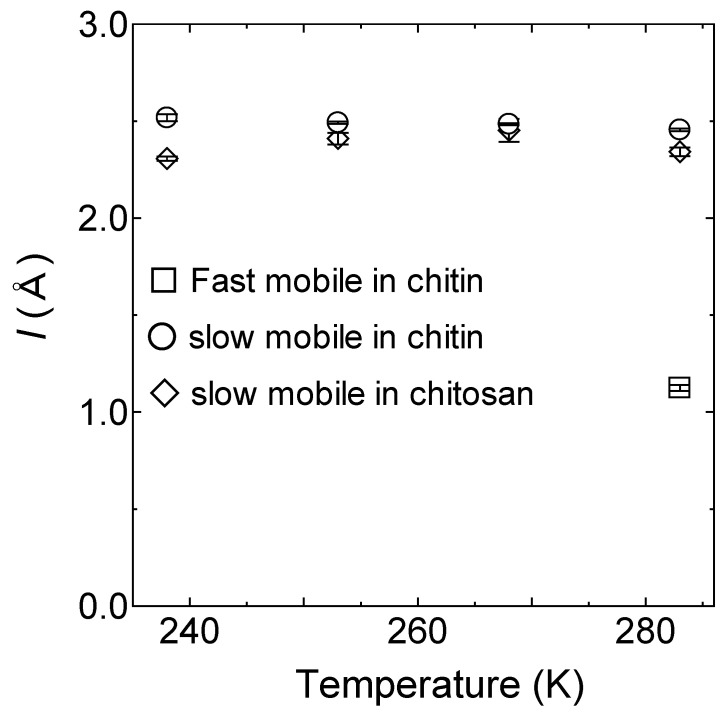
The jump distances *l* in chitin obtained from the QENS experiment based on *l* = $\sqrt{6D\tau }$ from 238 K to 283 K. The open circles and square are the slow and fast mobile hydration water in chitin, respectively. The open rhombuses are the slow mobile hydration water in chitosan [18]. The errors estimated from the fitting procedure are shown in a bar graph on each mark. Data in chitosan is adapted with permission from Ref. [18].

**Figure 9 bioengineering-10-00622-f009:**
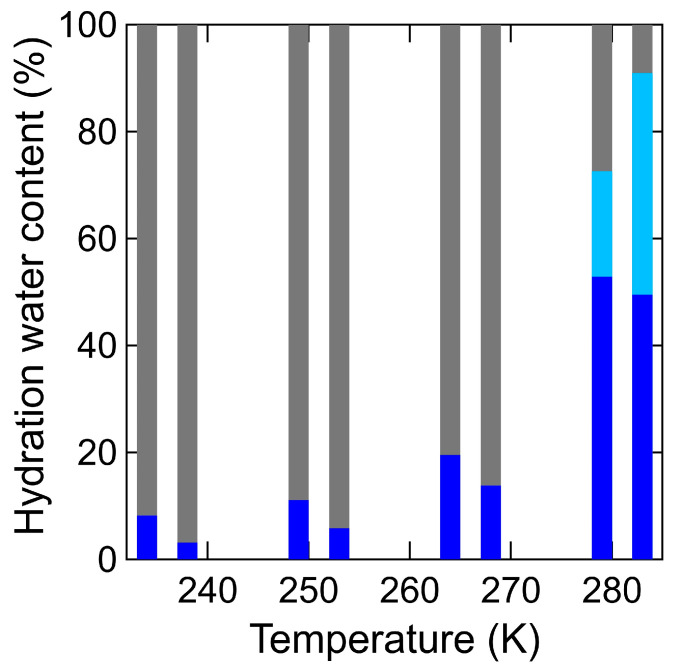
The temperature dependence of the fractions (*F%*) of three types for hydration water (i.e., the immobile, slow mobile, and fast mobile hydration water) of different mobility for the hydration water in chitin and chitosan. *F*% for the slow mobile (blue), fast mobile (light blue), and immobile (gray) hydrogen atoms from 238 K to 283 K in chitin (right) was estimated by Equations (4)–(6) and these of chitosan (left) [18]. Data in chitosan is adapted with permission from Ref. [18].

**Figure 10 bioengineering-10-00622-f010:**
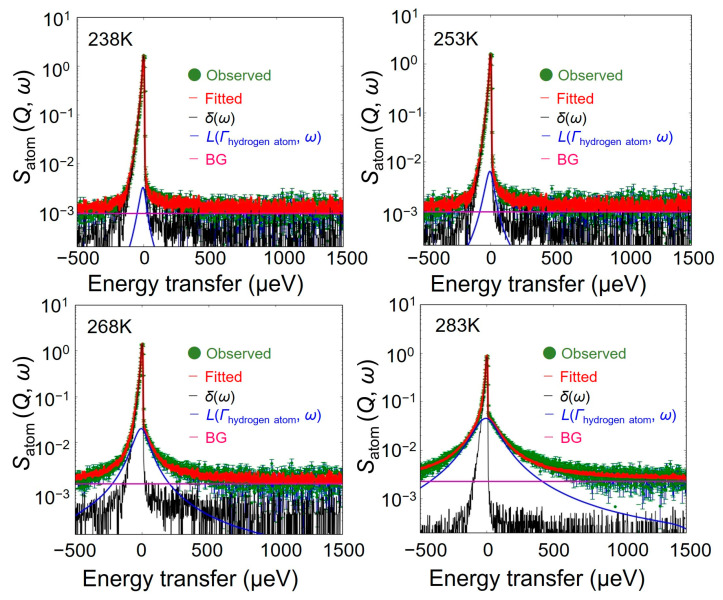
QENS spectra of *S*_atom_ (*Q*, *ω*) in chitin and fitting results from 238 K to 283 K at constant-*Q* = 1.0 Å^−1^. The green point and red line represent the experimental data and the results of fitting to Equation (7). The black and blue are *δ*(*ω*) and *L*(Γ_hydrogen atom_, *ω*) in Equation (7), respectively. These are convoluted with the instrument resolution. The magenta line is the instrumental background.

**Figure 11 bioengineering-10-00622-f011:**
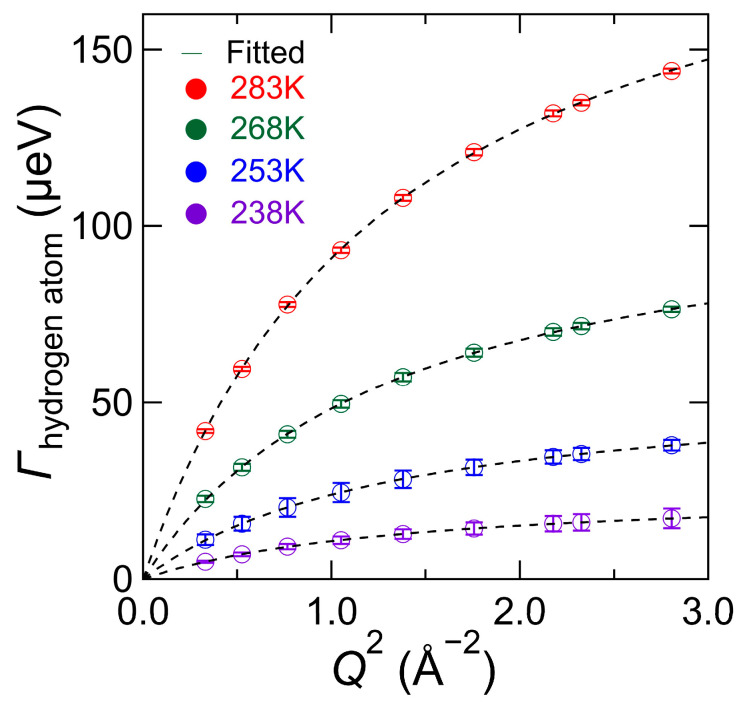
The *Q*^2^-dependence of half-width at half-maximum (HWHM) values of the quasi-elastic components (i.e., Γ_hydrogen atom_) for *S*_atom_ (*Q*, *ω*) in chitin obtained by QENS profile fitted to Equation (7). The experimental results at 238 K (purple), 253 K (blue), 268 K (green), and 283 K (red) for Γ_hydrogen atom_ are fitted by the jump-diffusion model (black broken line) [19].

**Figure 12 bioengineering-10-00622-f012:**
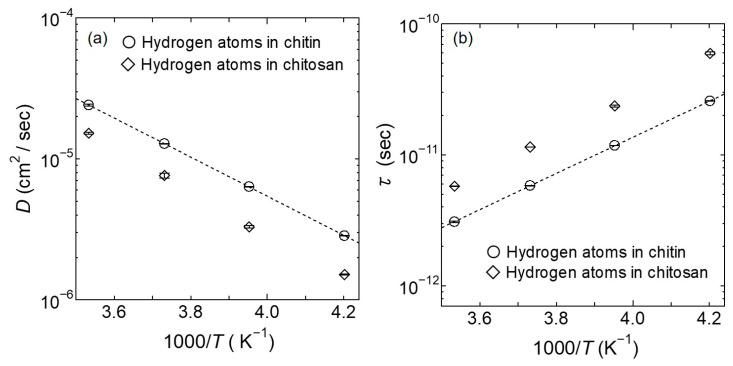
Arrhenius plot of (**a**) *D* and (**b**) *τ* for the mobile hydrogen atoms as a function of inverse temperature. The open circles represent *D* and *τ* results at each temperature for the mobile hydrogen atoms in chitin, and the open rhombuses are the hydrogen atoms in chitosan [18]. The black dotted lines on (**a**) *D* and (**b**) *τ* represent the Arrhenius fit to the data of the mobile hydrogen atoms in chitin. The errors estimated in the fitting procedure are shown in a bar graph on each mark. Data in chitosan is adapted with permission from Ref. [18].

**Figure 13 bioengineering-10-00622-f013:**
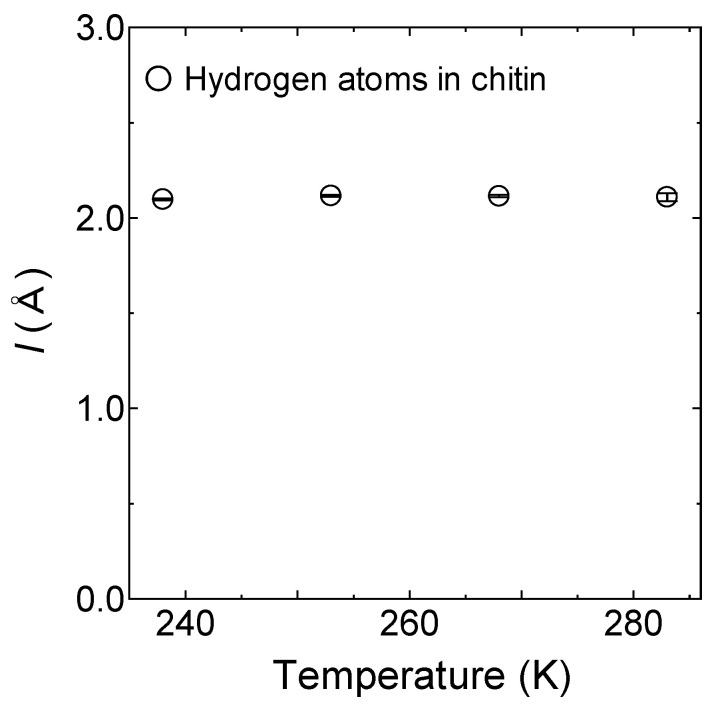
The jump distances in chitin obtained from the QENS experiment based on *l* = $\sqrt{6D\tau }$ from 238 K to 283 K. The open circles are the mobile hydrogen atoms in chitin. The errors estimated from the fitting procedure are shown in a bar graph on each mark.

**Figure 14 bioengineering-10-00622-f014:**
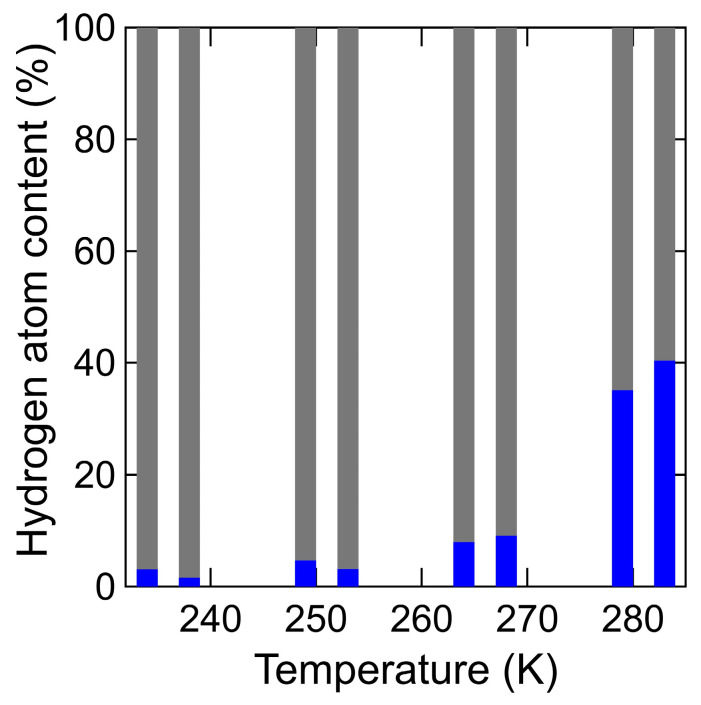
The temperature dependence of the fractions (*F%*) of two types for hydrogen atoms (i.e., the immobile, mobile hydrogen atoms) of different mobility in chitin and chitosan. (*F*%) between mobile (blue) and immobile (gray) hydrogen atoms in chitin (right) from 238 K to 283 K was estimated by Equations (4) and (5) and these of chitosan (left) [18]. Data in chitosan is adapted with permission from Ref. [18].

**Figure 15 bioengineering-10-00622-f015:**
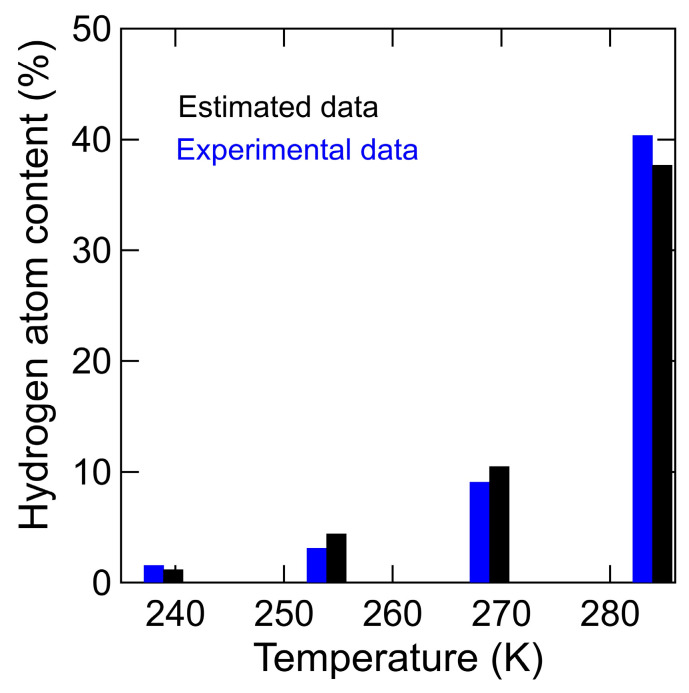
The fraction (*F*%) of the mobile hydrogen atoms in chitin between experimental (blue) and estimated data (black) from 238 K to 283 K. The experiment and the estimated data were obtained by Equation (5) and Equation (8), respectively.

**Figure 16 bioengineering-10-00622-f016:**
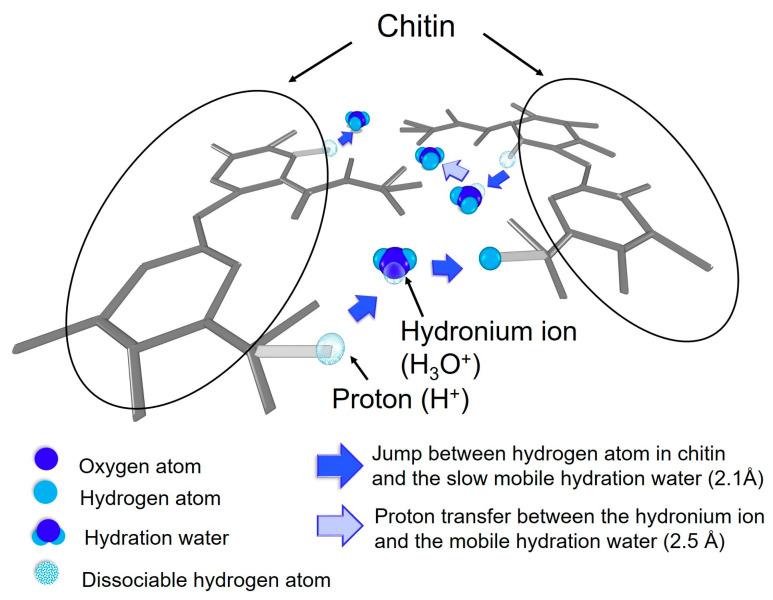
Diagrammatic proton conduction image in the hydrated chitin. In the first proton transfer process, the side chains of chitin (gray) with dissociable hydrogen atoms (the hydroxyl and N-acetyl aldehyde: light gray) dissociate as protons (translucent) and transfer to the neighboring slow mobile hydration water (oxygen: blue, hydrogen: light blue) to form H_3_O^+^. In the second proton transfer process, the hydrogen atom of H_3_O^+^ dissociates as a proton and transfers to another hydration water or to the proton accepters of the neighboring side chain of chitin. The proton conduction in the hydrated chitin is realized via these processes.

## Data Availability

The datasets generated and analyzed during the current study are available from the corresponding authors upon reasonable request.

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
