# Peer review of "Differences in Water Dynamics between the Hydrated Chitin and Hydrated Chitosan Determined by Quasi-Elastic Neutron Scattering"

_bioengineering, 2023, doi:10.3390/bioengineering10050622_

Round 1
Reviewer 1 Report
Yuki Hirota et al investigated the water dynamics between the hydrated chitin and hydrated chitosan determined by Quasi-Elastic Neutron Scattering technique. In fuel cells its really important to develop cheap electrolytic membranes to reduce the cost and alternative to Nafion membrane. These results indicate that the proton conduction process of the hydrated chitin is not only the proton transfer process between H3O+ and another hydration water as observed in the hydrated chitosan, but also the dissociable hydrogen atoms in the chitin preferetially transfer as protons to the slow mobile hydrogen water to form H3O+, and then the hydrogen atoms of H3O+ transfer to 3 proton acceptors of neighboring chitin.
While its great to investigate the water dynamics, I have only a single question. Why did authors not conducted the similar studies with the traditional Nafion membrane and compare the proton trsnfer mechanism of Nafion vs Chitin/chitosan membrane? Nafion membranes are considered a standard and any investigation should go through with Nafion membrane. Could you recommend which mechanism H+ ions tranfer in these membarnes? Growthus mechanism/hopping? Please comment and compare wherever, the superiority/inferiority of these membranes vs Nafion. The pictorial representation shown in figure 16 is very very poor please improve it.
other minor corrections
Line 50 --- fuel cell electrolyte where the reaction energy by hydrogen --->fuel cell electrolyte where the chemcial energy by hydrogen
Line 52 --- achieving zero emissions without waste or CO2 emission ---- What waste do you mean here?
All looks good
Reviewer 2 Report
The work continues a previous experiment to highlight the difference between water dynamics in hydrated chitin and hydrated chitosan using the quasi-elastic neutron scattering method.
General observation: In the figures, the font size is too large, and there is a contrast with the text of the work.
There may be some confusion in figure 1. How are the results in figure 1 obtained? They appear before the presentation of the experiment (Materials and methods).
The conclusions are supported by the experimental data. The references are balanced. I have not identified plagiarism or self-plagiarism.
There are no problems related to understanding the idea and the results of the experiment, except for the confusion mentioned above.
In my opinion, the work can be accepted for publication after eliminating the mentioned issues.
Reviewer 3 Report
In this manuscript, authors aiming the different proton conductivity between the hydrated chitin and hydrated chitosan, investigated the differences in water dynamics by quasi elastic neutron scattering. In general, the manuscript is well organized and it is an interesting work. However, there are still some issues to be addressed. A minor revision is required before its acceptance.
1. Why authors applied chitosan in this work should be further clarified with more detailed introduction on the structure, properties and applications of chitosan with necessary supporting articles: Recent advancements in applications of chitosan-based biomaterials for skin tissue engineering; etc.
2. Authors divided the introduction into many paragraphs, some of which could be merged to have a better story line.
3. The conclusion section can be shortened.
4. There are too many too old references, which is better to be deleted or replaced with recent articles to show the novelty of this work.
5. Authors should recheck the references to make sure full information is provided, such as volume, pages, etc. In addition, the format of references should be uniform.
Round 2
Reviewer 1 Report
Authors have satisfactorily addressed all the reviewers' comments and therefore, it can be accepted for publication.